# Application of Iron Nanoparticle-Based Materials in the Food Industry

**DOI:** 10.3390/ma16020780

**Published:** 2023-01-12

**Authors:** Dariusz Góral, Andrzej Marczuk, Małgorzata Góral-Kowalczyk, Iryna Koval, Dariusz Andrejko

**Affiliations:** 1Department of Biological Bases of Food and Feed Technologies, Faculty of Production Engineering, University of Life Sciences in Lublin, 20-612 Lublin, Poland; 2Department of Agricultural Forestry and Transport Machines, Faculty of Production Engineering, University of Life Sciences in Lublin, 20-950 Lublin, Poland; 3Department of Physical, Analytical and General Chemistry, Lviv Polytechnic National University, 79013 Lviv, Ukraine

**Keywords:** iron nanoparticles, food packaging, antimicrobial effects, enzymes, food analysis, food fortification, food storage, human health

## Abstract

Due to their different properties compared to other materials, nanoparticles of iron and iron oxides are increasingly used in the food industry. Food technologists have especially paid attention to their ease of separation by magnetic fields and biocompatibility. Unfortunately, the consumption of increasing amounts of nanoparticles has raised concerns about their biotoxicity. Hence, knowledge about the applicability of iron nanoparticle-based materials in the food industry is needed not only among scientists, but also among all individuals who are involved in food production. The first part of this article describes typical methods of obtaining iron nanoparticles using chemical synthesis and so-called green chemistry. The second part of this article describes the use of iron nanoparticles and iron nanoparticle-based materials for active packaging, including the ability to eliminate oxygen and antimicrobial activity. Then, the possibilities of using the magnetic properties of iron nano-oxides for enzyme immobilization, food analysis, protein purification and mycotoxin and histamine removal from food are described. Other described applications of materials based on iron nanoparticles are the production of artificial enzymes, process control, food fortification and preserving food in a supercooled state. The third part of the article analyzes the biocompatibility of iron nanoparticles, their impact on the human body and the safety of their use.

## 1. Introduction

Nanotechnology is one of the relatively younger fields of science. Thus far, it has been mainly associated with medicine, including drug delivery and pharmacotherapy. The development of nanotechnology in food is progressing much faster than the development of nanomaterials and nanopharmaceuticals. Industrialists, scientists and researchers are meeting the needs of these sectors with nanotechnology solutions. Nanotechnology is used in many sectors of the food industry, from agricultural production (herbicides, fertilizers, pathogen analysis and detection and genetic engineering) to crop processing and food production (intelligent packaging, improved taste, improved texture or quality of food, new gelling or viscosity control agents, dietary supplements) [1]. The properties of many materials depend on the size of the particles at the nanoscale, and their structure results in changes in the coercive force in magnetic materials, surface reactivity, catalytic ability or a very important feature, such as mechanical strength. The surface effects of nanoparticles are relevant to structural issues [2]. Compared to other metallic NPs, iron nanoparticles are widely used. This is determined by such properties as their small size, superparamagnetism and biocompatibility. Iron nanoparticles are used in bioprocessing, tissue engineering and other aspects of modern medicine. There are different oxide forms of iron, such as hematite (α-Fe_2_O_3_), maghemite (γ-Fe_2_O_3_), FeOH (OH) goethite and magnetite (Fe_3_O_4_) [3,4]. The most commonly used crystalline iron oxide structures are maghemite (γ-Fe_2_O_3_) and magnetite (Fe_3_O_4_). They are used in many fields, including magnetic data storage, pigment production for electrochemistry, contrast production and many others [5]. The antimicrobial, anti-larvicide and antioxidant activities of some iron nanoparticles are used in packaging applications. Iron oxide nanoparticles (IONPs) exhibit a number of beneficial features, such as magnetic properties, high values of saturation magnetization and control by low-intensity magnetic fields, as well as their non-toxicity, ease of biodegradation and compatibility with biological structures that are used in food production processes. Furthermore, the exploitation of the iron-binding properties of the molecule has been used to eliminate hazardous substances from food—for example, citrinin or the nephrotoxic mycotoxin that is found during fermentation and is produced by the *Monascus* mold. In the literature, IONPs are classified into the following categories: ultrasmall superparamagnetic iron oxides (USPIOs), cross-linked iron oxides (CLIOs) and monocrystalline iron oxide nanoparticles (MIONs) [1]

IONPs are being used to develop new proposals for biomedicine, food safety and environmental protection, as well as solutions without the limitations and drawbacks of currently available techniques. Among others, user-friendly, sensitive, low-cost diagnostic tests have been developed to assist in the detection of animal pathogens and diseases [6]. In addition, iron oxide nanoparticles offer the ability to bind to ligands, creating effective delivery tools. Thanks to this property, magnetic iron nanoparticles can be used in many areas of the food and medical industries, particularly as gene and drug nanocarriers [1].

The current emphasis on the use of nanomaterials is well justified, given the enormous potential that they exhibit. At the same time, their use in the food industry should be closely monitored. Over the past few decades, iron nanoparticles have gained a significant advantage in many technologies due to the many benefits associated with their use, including simple synthesis, ease of application and easy recovery through the use of magnetic fields. However, iron nanoparticles also have some disadvantages, such as aggregation and easy oxidation. This paper focuses on summarizing the applications of iron nanoparticle-based materials in food production and analyzes the safety of their consumption by humans in depth. In doing so, we provide evidence of the key role that iron nanoparticles will play in broadly understanding food production and highlight the health risks to consumers.

## 2. Methods for Obtaining Iron Nanoparticles

### 2.1. Chemical Synthesis

Many methods are known for the chemical synthesis of iron nanoparticles. Methods such as the co-precipitation of an aqueous solution of iron (Fe^2+^) and iron (Fe^3+^) salts by addition of a weak base (for example, sodium hydroxide, ammonia, TMAOH), microemulsion techniques, hydrothermal synthesis, sonochemical methods, non-aqueous methods, the thermal decomposition of an organic iron precursor, sol–gel techniques, γ-ray synthesis, microwave plasma synthesis and the thermal decomposition of an alkaline Fe^3+^ chelate solution in the presence of a high-boiling-point solvent have been described in the literature [7,8] (Table 1).

The most common methods of chemical synthesis are thermal decomposition and chemical reduction. Thermal decomposition methods proceed at elevated temperatures by using water or organic solvents. The process can be carried out at atmospheric pressure [33] or under increased pressure [34]. The size of the nanoparticles can be controlled by varying the nucleation rate. At higher temperatures, nucleation is faster than crystallization, so smaller nanoparticles are formed under these conditions [7]. The use of hydrophilic polyhydroxy alcohols as solvents facilitates the subsequent dispersion of nanoparticles in water [8]. Homogeneous iron nanoparticles of small size with high purity can be obtained by this method. Unfortunately, this comes at a high cost, as it requires high temperatures and expensive precursors that are hazardous to health and the environment. For this reason, this method is not suitable for industrial applications where large quantities of IONs are needed (e.g., environmental clean-up or Ni-Fe battery production). The use of toxic organic solvents, such as diethyl ether, eliminates the possibility of using IONPs in biological processes and food production [9,35]. Comparing the two methods of thermal decomposition and chemical reduction, it should be noted that chemical reduction is more economical. The greatest advantage of chemical reduction is that the nanoparticles produced can be biocompatible. In a reduction reaction, an iron salt, such as FeCl_3_, is reduced in an aqueous solution environment, resulting in the formation of IONPs. In the literature, the most commonly described reducing substances are hydrazine hydrate (N_2_H_4_·XH_2_O) and sodium borohydride (NaBH_4_). Hydrazine hydrate is used in alkaline environments, while sodium borohydride is used in neutral or acidic environments [36]. The easiest to carry out and most commonly used is the co-precipitation method. It involves the co-precipitation of a mixture in appropriate proportions of an aqueous solution of iron(II) and iron(III) salts (e.g., FeCl_3_·6H_2_O, FeSO_4_·7H_2_O) under alkaline conditions and in the absence of oxygen; this leads to the formation of nanocrystals of magnetite or maghemite. The size of the nanoparticles is manipulated by changing the pH or salt concentration of the solution [37]. However, the size of the nanoparticles is very uneven [11]. IONPs are most often produced without surfactants or polymers dissolved in water. These agents are only used to prevent aggregation and agglomeration. They cause the iron nanoparticles to repel each other [9].

### 2.2. Green Synthesis

#### 2.2.1. Synthesis by Microorganisms

Metallic and non-metallic nanoparticles are most often obtained by chemical processes carried out under rigorously controlled conditions. In contrast, the production of nanoparticles under biological conditions is characterized by such conditions of temperature, pH or pressure, which can be easily obtained both in the laboratory and in industry [2]. In living organisms, the intracellular or extracellular generation and deposition of inorganic crystalline structures is common. Examples include CaCO_3_ and Ca_3_(PO_4_)_2_ deposited in bacterial structures; gold and silver deposited in mycelia and bacteria; and silica, magnetite and greigite found in magnetotactic bacteria [8,37]. The potential of microorganisms is increasingly being exploited in laboratory conditions for the preparation of iron nanoparticles (Table 2).

The combination of microorganisms with metal ion precursors leads to products that possess great potential for use in nanoparticle synthesis.

#### 2.2.2. Synthesis Using Plant Parts

The simplest, cheapest and most reproducible means to produce biocompatible IONPs is green synthesis. Different parts of plants (leaves, stems, seeds and roots) are used for this purpose (Table 3).

The ease of synthesis of nanoparticles in biological conditions is caused by the high content of organic reducing complexes contained in plants. Plants, compared to microorganisms, synthesize stable nanoparticles that are useful for rapid, large-scale synthesis [66].

Many studies have investigated green plant parts for the environmentally safe production of iron nanoparticles using precursors such as iron(II) chloride or iron(III) chloride solution. This has been shown to be possible due to the high content of bioactive components in plants. These active substances are mainly polyphenols, which have reducing and encapsulating properties. In addition, the use of a water environment and normal pressure and temperature eliminates the possibility of the formation of toxic substances, which in turn results in a lack of problems regarding the disposal of hazardous waste. For example, citrus and peanut peels and stalks can be used in nanoparticle production. Chemical and physical methods of producing nanomolecules require the usage of very reactive and toxic reducing agents, e.g., sodium borohydride and hydrazine hydrate, that cause unwanted effects to the environment, plants and animals. Such agents are not used in green synthesis [67].

#### 2.2.3. Synthesis Using Green Reagents

There have been many studies that have used non-toxic, biocompatible materials to produce IONPs. Analyses have been conducted using biopolymers, ascorbic and amino acid, hemoglobin and myoglobin, sugar, glucose, synthetic tannic and gallic acid, among others [66]. Examples of green reagent applications are presented in Table 4.

## 3. Application of Iron Nanoparticle-Based Materials in Food Production

### 3.1. Application in the Production of Food Packaging

The use of nanomaterials in food packaging alters the penetration properties, improves the barrier properties, increases thermal resistance and leads to strong antimicrobial activity [77]. Nanoparticles are characterized by their large surface area and high surface energy. This creates strong interfacial interactions between the polymer bonds and the nanoparticles. This significantly improves the properties of the bio/polymers used for packaging [78].

One of the main problems during food storage is its contact with oxygen. This agent is responsible for many degradation reactions in food, such as lipid oxidation, microbial increase, enzymatic browning and nutritional degradation. Changes in food as a result of oxygen exposure result in a significantly reduced shelf life, especially for oxygen-sensitive foods [79]. In order to limit the access of oxygen to food, it is most often packaged using materials that provide an active barrier against oxygen penetration [80]. One way in which food can be protected from oxygen is through the use of iron nanoparticles. Two strategies are possible for the use of nano-iron in packaging. The first is to insert sachets containing nZVI into the packaging, and the second method is to incorporate the nanoparticles into the packaging film using a monolayer or multilayer structure [81,82]. An example is bentonite and kaolinite modified with Fe^0^. These substances have been approved by the European Food Safety Authority (EFSA) as non-nanoparticles [83]. Unlike normal iron, Fe^0^ reacts with oxygen under non-humid conditions [80]. Khalay et al. [84] investigated the suitability of PP nanocomposites having in their compositions montmorillonite (OMMT) with IONP modification for food packaging. They found that the nanoparticles were active as an active oxygen scavenger, and were able to capture and absorb oxygen by chemically reacting with it. Similarly, Busolo and Lagaron [82] reported that iron kaolinite contained in active composites is a passive barrier due to its winding pathway, which impedes gas diffusion and captures and reacts with molecular oxygen.

The most important problem during food storage is spoilage by microorganisms. The use of IONPs and nanocomposites with antimicrobial functions is an effective means to minimize the impact of microorganisms on food during processing or storage. In this way, they provide an extended shelf life and improved food safety [85]. Iron nanoparticles can be used to address food oxidation by applying antimicrobial coatings to the inner surface of the packaging. Such a coating gradually releases the antimicrobial substances, or they are immobilized on the active surface of the packaging [86]. Song et al. [87] produced composites of polydopamine with iron oxide nanoparticles (IONPs@pDA). Nisin was conjugated on the IONP@pDA nanoparticles. The newly synthesized material showed good efficiency in reducing Alicyclobacillus acidoterrestris, a food-spoilage-causing bacterium that is a serious problem in the food industry. Recent research has focused on producing IONP composites with surfaces modified with other nanometals. There is strong interest in silver-coated magnetic nanocomposites because of their unique antimicrobial properties [88].

### 3.2. Edible Coatings on Food

The use of films and coatings produced from natural edible polymers (biopolymers) on food is increasingly being used to extend the shelf life of fresh produce. These films and coatings have similar functions to conventional protective packaging or synthetic coatings [89]. The use of food-approved nanoparticles has enabled the creation of functional edible coatings. These coatings consist of nanoemulsions, polymer nanoparticles, nanofibers, solid lipid nanoparticles (SLNs), nanostructured lipid carriers (NLCs) and traditional polymer nanocomposites (inorganic–organic) [90]. One of the most commonly used ingredients in edible films is chitosan. Chitosan is a natural polymer composed of (1,4)-linked 2-amino-deoxy-b-D-glucan. It has been shown in many publications that chitosan is non-toxic, biodegradable, biofunctional, biocompatible and has antimicrobial properties. Furthermore, due to its ability to form transparent films and the good mechanical properties of films that can meet various packaging needs, chitosan is an excellent ingredient for edible packaging [91]. Combining chitosan with nanoparticles seems to be an excellent solution. Nehra et al. [92] reported that chitosan coated with Fe_3_O_4_ nanoparticles is an antimicrobial agent against *E. coli*, *B. subtilis*, *Candida albicans*, *Aspergillus niger* and *Fusarium solani*. Shrifian-Esfahni et al. [93] found that under in vitro conditions, a chitosan–Fe_3_O_4_ nanoparticle film showed inhibitory activity against *E. coli* and *P. aeruginosa*. SPIONs composed of magnetite (Fe_3_O_4_) are characterized by such properties, which can be used for antimicrobial applications [94], especially since the U.S. Food and Drug Administration has found that SPIONPs are biocompatible (BC) with the human body [95].

### 3.3. Immobilization of Enzymes

Enzymes are used as biocatalysts for large-scale production in many areas of industry. Their great advantage is their limited environmental impact. In the food industry, they make it possible to modify the structures of foods. For example, this is used in the production of lactose-free milk and calorie-reduced fats [96]. Placing enzymes on magnetic nanoparticle carriers ensures a strong covalent bond between the enzyme and the carrier. By doing so, it minimizes the leaching of the enzyme in the aqueous environment and prevents the contamination of the product with other proteins. An important advantage of this solution is that a magnetic field can be used to separate the biocatalyst from the reactants. This protects against particle losses and provides high mechanical strength, thermal stability and resistance to chemical and microbial degradation [97]. Another area where iron SPIONs have found application is in protein/enzyme immobilization. Here, the characteristic magnetic properties of SPIONs and the ease of separation in a magnetic field have been exploited. However, the most important factor is the ability to orient proteins/enzymes in the carriers; this protects against internal diffusion, which is a problem in traditional methods [98].

### 3.4. Artificial Enzymes

Nanomaterials that exhibit enzyme-like activity are proposed as a new generation of artificial enzymes and referred to as “nanozymes” [99]. The main advantage of nanozymes over their natural counterparts is the ability to regulate activity by changing the size or morphology, or through doping or surface modification. The most typical nanozymes are iron oxide nanoparticles (Fe_3_O_4_ and Fe_2_O_3_). They mimic the activity of peroxidase and catalase [100]. Peroxidase-like activity is possible under acidic conditions. Ferromagnetic nanoparticles can catalyze the reaction of hydrogen peroxide with hydroxyl radicals. This means that IONPs can be referred to as IONzymes. As reported by Qin et al. [101], IONzymes are antiviral agents because they induce the peroxidation of membrane lipids in synthesized liposomes. Their large surface area enables them to exert strong catalytic activity. This activity matches and sometimes exceeds that of natural enzymes. IONzymes are more stable and less expensive than natural enzymes. In addition, proteolytic degradation and environmental factors (temperature, pH, ionic strength and heavy metals) are no longer an issue. One of the most important features is that they do not degrade during storage at room temperature. The disadvantages of IONzymes are that they only bind to a specific substrate (substrate specificity) and show different catalytic activity, which is determined by the size and structure [102].

### 3.5. Food Analysis

Iron nanoparticles have been widely used in food analysis methods [103]. Magnetic nanoparticles are usually integrated into detection techniques in two ways: as an electrode modifier and as a sample preconcentrator [98]. An example is the method based on magnetic microparticles coated with protein A (a component of the cell wall produced by several strains of *Staphylococcus aureus*). This has been used for the accurate (0.2 mg/kg) detection of the allergen Ara h3 (a heat-stable protein that comprises 19% of the total protein in peanut extracts) in food. The method has high repeatability and reproducibility [104]. Another example is the use of magnetic nanoparticles in glucose biosensors. The biosensors use immobilized oxidase to convert the target analytes into electrochemically detectable products. This makes the determination of glucose in food easier [105]. Magnetite nanoparticles enable food analysis for the detection of certain elements (e.g., silver, copper, lead, cadmium and mercury). Magnetite nanoparticles are coated with silanes. This allows them to be functionalized with ligands that are selective for metal ions [103] (Table 5).

In a study by Yang et al. [117], it was shown that SPIONs can accelerate the preconcentration process for the detection of bacteria in food. Target bacteria used for PCR detection are concentrated using submicron superparamagnetic anion exchangers. This enabled the PCR detection limit to be shifted from 10^4^–10^5^ CFU mL^−1^ to 10^2^ CFU mL^−1^. *Escherichia coli* and *Agrobacterium tumefaciens* were used as models of bacteria. Wen et al. [115] described a one-step assay for the detection *of Salmonella typhimurium* by fluorescent identification using IONPs with functional additives. The work of Chen et al. in [118] used silica-coated SPIONs to extract DNA from pathogenic bacteria. In this method, preincubation of the sample was avoided and the influence of the food matrix was eliminated.

### 3.6. Protein Purification

One increasingly used process in the food industry is the purification of proteins from isolates or concentrates. The resulting proteins can then provide the basis for new forms of food. For example, milk proteins fall into two distinct categories: casein micelles, which are often associated with cheese production, and serum proteins, which are involved in many functional properties, such as foaming, gelling and emulsification. In high-tech food production, it is necessary to use only one type of protein. Therefore, there is a need for a low-cost and efficient technique to obtain pure proteins [119]. Traditionally, proteins are purified by chromatography, precipitation, ultrafiltration, centrifugation and dialysis. The inherent limitations of these methods are the long pretreatment times, equipment that is often expensive to purchase and operate and the need to hire experienced lab technicians. Magnetic separation, which has been known for many years, can be used for this purpose. It is characterized by several advantages. The technique using SPIONs is fast, scalable, sensitive and easy to automate and no special sample preparation is needed (Figure 1).

SPIONs used for protein purification are immobilized with ligands. Primarily, these are ligands with confirmed affinity for the proteins to be purified [105].

Another example is the use of SPIONs for the analysis and separation of hormones, drugs and contaminants from food. This technique is used in the study of food of animal origin. Traces of drugs may be present in this type of product. Such contaminated products, when consumed by humans, endanger their health [103]. Estrogen hormones are administered as feed additives (called growth promoters) to accelerate growth and increase meat weight in cattle, poultry and pigs [120]. As reported by Gao et al. [100], functionalized Fe_3_O_4_ nanoparticles have been successfully used as sorbents for the specific separation and enrichment of proteins from milk. The authors report that they were able to recover estradiol in the range of 88.9 to 92.1%. Luo et al. [121] used carbon nanotubes with SPIONs because of their excellent adsorption capacity against hydrophobic compounds. They were used as adsorbents in the extraction of phthalate acid esters. Esters such as phthalates and bisphenol A are derived from plastic packaging and are considered as substances with potential carcinogenic properties.

### 3.7. Iron Oxides as Ingredients in Foods and Dietary Supplements

The most important biological function of iron is oxygen transport, as it is part of the heme nucleus in the proteins hemoglobin and myoglobin. In living organisms, iron is involved in more than 200 enzyme systems that are essential for cellular function, particularly cellular energy utilization and DNA, RNA and protein synthesis. In addition, iron is involved in neurotransmitter metabolism, vitamin D activation, collagen metabolism and cholesterol catabolism. Therefore, iron is essential for the proper functioning of organisms, including humans [122]. According to Dave and Gao [33], ferritin is an intracellular protein found in the cells of the liver and immune system. It stores inactive iron in the body and releases it when needed. As reported by Kumari and Chauhan [123], commonly used iron enrichment additives with high bioavailability (FeSO_4_) are chemically reactive. These reactions cause color changes, a metallic aftertaste, rancidity in food products, etc. Due to these difficulties, FeSO_4_ is replaced by water-insoluble compounds FePO_4_, Fe_4_(P_2_O_7_)_3_ or ions of iron. These additives do not affect the properties of food but are poorly absorbed by the human body [124]. Therefore, IONPs are an alternative to traditional iron supplements, preventing iron-deficiency-related diseases [125], as similarly stated by von Moos et al. [124] and Perfecto et al. [126], using FePO_4_ NPs as an iron source. In addition to iron oxides and phosphates, Hilty et al. [127] used a Ca and Mg complex in iron oxide compounds. These minerals are essential for humans, so adding them to food poses no health risk. The authors showed that the addition of calcium and magnesium to Fe_2_O_3_ gives i.d.a. solubility > 85%, suggesting a high level of bioavailability. In addition, no caking was found during storage under various humidity conditions.

### 3.8. Colorants

Iron, in the form of various compounds, is used as a colorant (food additive E172). A range of colors can be obtained, such as black, red, yellow, blue, orange and brown, depending on the chemical composition. For example, magnetite (Fe_3_O_4_), with a particle size of 11.7 nm, is characterized by its black color. The red pigment is associated with hematite (α-Fe_2_O_3_) nanoparticles with a particle size of 39.5 nm, and the yellow pigment is attributed to goethite (α-FeOOH) nanoparticles with a particle size of 48.7 nm [128]. Iron oxides are approved food pigments and have been shown to contain significant amounts of nanoparticles [129].

### 3.9. Mycotoxin Removal

Mycotoxins are produced by filamentous fungi belonging to the Ascomycota cluster or molds and are toxic secondary metabolites. Their characteristic feature is their low molecular weight. The source of mycotoxins is usually infected food products. They are often carcinogenic and mutagenic; among other aspects, they inhibit DNA synthesis and cause changes in RNA metabolism [130]. During various food processing technologies, including cooking, baking, frying, roasting and pasteurization, most mycotoxins remain chemically and thermally stable. Therefore, low-cost methods to eliminate these contaminants from food are still being sought [131]. Iron oxide and graphene oxide nanostructures can be cheaply produced and are readily available. Mycotoxins interact with the surface oxygen functional groups of the nanostructures. Surface-active maghemite nanoparticles (γ-Fe_2_O_3_) have shown chelating properties for citrulline and ochratoxin A in the presence of iron(III). Another promising solution is IONP@chitosan complexes as patulin adsorbents. As reported Horky et al. [132], after approximately 5 h, the patulin molecules were completely adsorbed. The adsorbent concentration was 400 μg, with a pH equal to the pH of apple juice. After adsorption, mycotoxins along with SPIONPs were removed in a magnetic field.

### 3.10. Anti-Allergic Effect

Histamine is found in some foods. In food, histamine is formed during storage as a result of the activity of bacteria—not only those intentionally added, but also those that contaminate it. Histamine content is considered one of the markers of food quality [133]. This compound is one of the most important intermediates in the course of allergic reactions. Consumption of histamine-rich foods can cause nausea, headaches, diarrhea and asthma [134]. According to Ghanbari, Adivi and Hashemi [134], magnetic Fe_3_O_4_@ agarose/IDA@silica nanoparticles can be used to remove histamine from biological fluids. The histamine removal efficiency of real plasma samples was 92%.

### 3.11. Control of the Process Flow

Bacillus subtilis is used for the industrial production of a wide range of compounds in the fermentation process. One of the main problems of industrial fermentation is the formation of biofilms by this bacterium, which leads to many process and operational complications. Biofilms are an unfavorable phenomenon in industrial settings. Periodic cleaning of equipment is necessary. The use of SPIONs coated with aminopropyltriethoxysilane is a promising approach to control biofilm formation without losing the viability of B. subtilis cells [135].

### 3.12. Preservation of Food in Supercooled State

Iron nanocrystals (Fe_3_O_4_) are a natural component of meat. They provide strong nucleation sites for heterogeneous ice crystallization. The ferromagnetism of magnetite nanoparticles allows an external oscillating magnetic field to induce magneto-mechanical motions. These movements can interfere with the formation of ice crystals in food by inducing a supercooled state. For this to occur, the magneto-mechanical rotation of the particles should be greater than the magnitude of Brownian motion [136]. A study by Kang at al. [137] found that fresh beef stored under refrigeration at −4 °C using an oscillating magnetic field was preserved for one week longer, without significant changes in weight loss, color or bacterial growth.

## 4. Effects of Iron Nanoparticles on Living Organisms

### 4.1. Antimicrobial Activity

A number of papers have shown that certain metal nanoparticles exhibit antimicrobial activity against a variety of Gram-negative and Gram-positive bacteria [8] (Table 6).

The most common action of NPs is to damage bacterial cells by destroying the cell membrane, damaging organelles, biomolecular distortion and by interfering with the biosynthesis of nucleic acids and proteins [8].

Damage to nucleic acids and proteins in a bacterial cell is caused by superoxide radicals (O_2_^−^), hydroxyl radicals (-OH), hydrogen peroxide (H_2_O_2_) and singlet oxygen (O_2_). These are referred to as reactive oxygen species (ROS) [149]. Proteins including enzymes are inactivated or the reaction is caused to slow down by partial inhibition with metal ions capable of binding to mecapto (-SH), amine (-NH) and carboxyl (-COOH) groups. In addition, IONPs are able to interfere with F0F1-ATPase/ATP synthase function and reduce the rate of H+ flow across the membrane and the redox potential [150]. Lee et al. [10] showed that Fe^0^ nanoparticles bind to intracellular oxygen. This phenomenon causes oxidative stress and the breakdown of the cell membrane. In addition, they found that the strong antibacterial activity allows the use of nano-Fe^0^ as a biocide in many applications, also in combination with silver nanoparticles. The main problem is the strong oxidation of nZVI, which results in a reduction in their bactericidal activity, thus limiting their application. Therefore, work is needed to obtain nanoparticles that are resistant to oxidative corrosion.

Another means of exploiting the antibacterial properties of iron nanoparticles is through the catalytic reaction of Fe_3_O_4_ with H_2_O_2._ The reaction produces free radicals (intermediate products). These radicals are very toxic to microorganism cells, because they attack membranes, proteins and nucleic acids and ultimately cause the bacteria to malfunction [100]. Gabrielyan et al. [140] studied the impact of Fe_3_O_4_ on selected strains of Gram-positive and Gram-negative bacteria. They found that Fe_3_O_4_ was more effective against Gram-negative *E. coli* than Gram-positive *E. hirae*. The results prove that membrane permeability can be modified in the presence of Fe_3_O_4_.

Another use for iron nanoparticles is their ability to induce hyperthermia, a state of elevated temperature caused by an alternating magnetic field [16]. Superparamagnetic iron oxide is used to inactivate bacteria by creating elevated temperatures [142]. According to Rodrigues et al. [143], hyperthermia allowed the complete elimination of *P. fluorescens* at the temperature of 50–55 °C.

### 4.2. Antiviral Effect of IONPs

The antiviral activities of NPs occur both inside and outside cells. They involve NPs interacting with gp120-type proteins, competing with the virus for binding sites to the host cell, inferring viral attachment and preventing binding or blocking viral penetration into the host. Further mechanisms of antiviral action include acting on viral particles, disrupting the viral genome or binding to viral particles, and thereby preventing viral replication [151]. For example, Fe_3_O_4_ nanoparticles inhibit DEN-2 virus replication by inhibiting E protein expression [141]. Another action of iron NPs is to bind to the virus to prevent it from binding to cells.

### 4.3. Antifungal Activity

Iron oxide nanomaterials can potentially be used against many fungal pathogens (Table 6).

The small size of the nanoparticles and their greater surface area to volume ratio is the main mechanism of antimicrobial activity. This mechanism effectively weakens the microorganisms and reduces the supply of oxygen for respiration [146]. In addition, the same effects as in bacterial cells are displayed; i.e., IONPs are able to depolarize the cell membrane to induce the production of ROS and generate oxidative stress that disrupts cellular homeostasis [152].

### 4.4. Effects of Iron Nanoparticles on the Human Body

Iron nanoparticles can be toxic. Their intracellular and in vivo toxicity results from the production of excess reactive oxygen species (ROS), including free radicals such as superoxide anion, hydroxyl radical and non-radical hydrogen peroxide. High levels of free radicals can damage cells by peroxidizing lipids, damaging DNA, modulating gene transcription and altering proteins, and can cause impaired physiological functions and cell apoptosis. There are four main sources of oxidative stress in response to iron oxide nanoparticles (Figure 2):Direct generation of ROS on the surfaces of nanoparticles;Production of ROS via the leaching of iron particles;Altered function of mitochondria and other organelles;Induction of cellular signaling pathways [139].

### 4.5. In Vitro Evaluation

Cell viability analysis, proliferation, differentiation using atomic force microscopy and gene expression analysis are commonly used in vitro toxicity testing techniques [153,154]. An initial assessment of biocompatibility is achieved by examining all of these parameters. This is important in determining the interactions of nanoparticles with the cell membrane, aggregation and the presence of physiological effects. Evaluating the safety of nanoparticles in vitro is easy and inexpensive. Nanoparticles can affect the proper functioning of cellular structures (e.g., ROS generated by nanoparticles increase the activity of mitochondrial enzymes, reactions between some dyes used in experiments and nanoparticles), which should be taken into account when conducting experiments [155].

### 4.6. In Vivo Evaluation

The interactions of magnetic nanoparticles in biological systems are highly dynamic and complex. Investigation of the influence of nanoparticles on cellular function is important for many reasons. An important phenomenon is how IONPs affect physiological iron metabolism after they are degraded in the body. The accumulation of intracellular iron can affect the denaturation of proteins and nucleic acids [156].

### 4.7. Biocompatibility

Biocompatibility primarily depends on the degree of observed cytotoxicity. However, there are no clearly defined criteria for NP cytotoxicity. In recent studies, it has been observed that they can cause DNA damage, mitochondrial membrane dysfunction, oxidative stress and changes in gene expression. However, cytotoxicity associated with the above problems was not observed [157]. Biocompatibility is a requirement in classifying nanoparticles including iron for biomedical applications. Confirmation of biocompatibility is achieved by performing in vitro and in vivo analyses. Cytotoxicity investigations mainly include cell viability/proliferation/differentiation tests, microscopic analysis and intracellular localization, hemolysis and genotoxicity tests. In order to provide information on the effects of nanoparticles on living organisms, including blood compatibility, biodistribution, metabolism and clearance, studies should be performed both in vitro and in vivo [158]. It is widely believed that iron nanoparticles are the most biocompatible among other nanosubstances. However, in scientific works, one can find different opinions on this issue. In general, it has been shown that the toxicity of nanoparticles depends on their size, surface properties, concentration, shape and structure [159,160]. For this purpose, a number of in vitro biocompatibility studies of nanoparticles of various shapes (including nanooctahedra, nanorods, nanocubes, nanohexagons, nanodrills, nanotubes, nanobeads, nanobeads, nanorods, nanorings and nanocapsules) were carried out. Zhou et al. [161] conducted in vitro analyses of differentiated IONPs (nanooctahedrons, nanorods and nanocubes) in A549 lung cancer cells. More than 90% of the cells survived after 24 h of incubation. This proved that the nanoparticles were safe for cells in a given range (10–1000 μg mL^−1^). Chu et al. [162] studied the effects of IONPs of different shapes coated with poly(ethylene glycol)-phospholipid (DSPE-PEG) on ECA-10^9^ cells. Laser irradiation of nanoparticles caused negligible toxicity to cells. On the other hand, Fan et al. [163] analyzed the cytotoxicity of magnetite nanoparticles. The viability results showed that ferrimagnetic IONP nanorods exhibited negligible toxicity at low IONP concentrations (50 μg mL^−1^ of Fe_3_O_4_), after 24 h of incubation. According to these authors, the shape of the nanoparticles is not as important as the concentration. The concentration of iron nanoparticles is a key factor in assessing cell viability during incubation. Ankamwar et al. [159] observed the cytotoxicity of IONPs at concentrations above 100 μg mL^−1^. Nanoparticle size is an important factor in biocompatibility. To increase the filtration time through the spleen and liver, nanoparticles should be small in size (<200 nm). On the other hand, to avoid rapid flow through the kidneys, the size should be larger than 10 nm [164]. An important property of nanoparticles that affects the human body is their surface area. Uncoated IONPs tend to be more toxic in vitro as well as in vivo compared to encapsulated nanoparticles. In medical applications, surface coating of IONPs is often used. In addition, it can protect against the agglomeration of IONPs and improve dispersion, support chemical drug combination modes and reduce non-specific effects on cells, providing lower toxicity and higher biocompatibility [165]. For obvious reasons, the coating materials used should have high biocompatibility and affinity for IONPs and should not affect the immune system or be antigenic. The most commonly used coatings are composed of various materials, both organic and inorganic [166]. One type of material used to protect iron nanoparticles is polymers, such as polyethylene glycol (PEG), which has been approved by the US Food and Drug Administration (FDA) and is considered a safe and biocompatible material. The advantage of this material that it is easily removed by the kidneys (for PEG < 30 kDa) or the digestive system (for PEG > 20 kDa). In addition, the adhesion of PEG particles to the surfaces of iron nanoparticles allows for an extended residence time in the bloodstream [167]. Other materials used to enhance the biocompatibility of IONPs include polyvinyl alcohol (PVA), polyethyleneimine (PEI), polyacrylic acid (PAA), poly(lactide-co-glycolide) (PLGA), dextran and protein molecules [138,168,169,170,171,172]. Disadvantages associated with the use of polymers include the in vivo aggregation of nanoparticles and the ease of separation of the colloidal suspension [159]. In a study by Han and Zhou [173], the enhanced biocompatibility of liposome-modified surface nanoparticles was demonstrated over an appropriate concentration range.

### 4.8. Effects on the Human Body

Iron nanoparticles have broad effects on the human body. Fe_2_O_3_ NPs have been found to protect the heart from ischemia at the tissue and cellular levels in an ischemia–reperfusion model. The therapeutic effect on iron deficiency occurs at doses of 3 mg to 6 mg per kilogram of body weight per day. Exceeding the recommended standard (10–20 mg/kg body weight) causes adverse effects [86]. One of the potential mechanisms of IONPs is to inhibit intracellular ROS and reduce peroxidative damage [174]. More interestingly, Zhang et al. [175] described that the consumption of IONPs can result in slower aging and prevent neurodegeneration. Feitosa et al. [176] found that oxidative stress causes aging associated with slowed cellular metabolism by measuring the clonogenic potential of LA-9 fibroblasts at day 7 after exposure to Fe_3_O_4_ NPs. Similarly, Fe_3_O_4_ nanoparticles modified with poly(L-lysine) can promote the proliferation of cancer stem cells from U251 glioblastoma multiforme via a reduction in the level of intracellular H_2_O_2_ [177]. In studies on aggregation into larger clusters of protein amyloid (Parkinson’s disease, Alzheimer’s disease and type II diabetes), it was shown that the amount of protein aggregates was decreased by the use of Fe_3_O_4_. IONPs inhibited the aggregation of lysozyme amyloid by blocking nucleation and caused the depolymerization of lysozyme clusters by interacting with adjacent protein sheets and breaking them down [178]. High concentrations of iron in the body can be toxic and lead to abnormalities in tissue homeostasis. It can cause oxidative stress, cytotoxicity, DNA damage, epigenetic events and inflammatory processes. Even if the high iron concentration of iron ions does not cause cytotoxicity, disturbances in cellular structures may occur, initiating carcinogenesis. Some studies have shown that iron may be responsible for the occurrence of cancer formation by, for example, generating ROS, which can damage DNA and proteins and cause lipid peroxidation [157]. However, numerous works confirm that IONPs are safe and non-cytotoxic at concentrations up to 100 μg mL^−1^ [156,159,179]. Before administering nanoparticles, there are different paths for their absorption, such as digestion, inhalation or dermal absorption.

## 5. Summary

Iron nanoparticles are increasingly used in food applications. This is influenced by their low manufacturing cost, the possibility of using their properties in a wide range and their biocompatibility. This review highlights the most important directions of nanoparticle use in the food industry. This is influenced by their high reactivity with oxygen, both in the presence of moisture and in anhydrous environments, and their ability to counteract the growth of microorganisms belonging to different groups in terms of morphological structure (Gram-negative and Gram-positive bacteria, yeasts and molds). The use of iron nanoparticles in the form of ZVI is particularly important. Iron nanoparticles, due to their high surface-area-to-size ratio, have become versatile tools in terms of creating carriers to stabilize enzymes. Enzymes that have been previously immobilized have greater resistance to changes in environmental conditions (pH, temperature) than enzymes not treated with this technique. In addition, they can be reused because of their ease of recovery. There are also disadvantages to this technique. The activity of some enzymes decreases, indicating the need for further research on this technique. The results obtained in the separation or purification of proteins from solutions through the use of SPIONs indicate that it is possible to transfer this technique to the removal of some proteins from food. Due to the magnetic properties of SPIONs, it is also possible to eliminate many adverse components, both pathogenic and allergenic, from food. Magnetic SPIONPs integrated with PCR, immunoenzymes, spectrometry and biosensors are fast and accurate methods for the detection of certain ingredients or contaminants in food, and they are helpful in identifying and quantifying microorganisms. The use of SPIONs increases the sensitivity and speed of analysis. It can also reduce the cost of manufacturing equipment. Another promising application of SPIONs is that they can be used to control the ice nucleation process in the freezing and storing of food.

Another direction of application for iron nanoparticle-based materials in the food industry is the fortification of food products. Iron at nano-size has high bioavailability, and problems such as an unacceptable color, taste, metallic aftertaste and rancidity in foods due to the use of water-soluble iron salts have been solved.

On the other hand, many papers have been written on the possible toxic effects of IONPs on the human body. The small size of the nanoparticles allows them to pass through physiological barriers, by which they can adversely affect cells or tissues, consequently leading to health risks. The coating of IONPs and their size are key to inducing toxicity mechanisms. Therefore, in order to safely use iron nanoparticle-based materials in the food industry, the safety and effects of IONPs on the human body must always be evaluated.

In conclusion, iron nanoparticles can be successfully used for the production of packaging that comes into contact with food, as well as the food itself. Smaller dimensions mean a larger external surface, which in turn improves water absorption, the release of aromas, the availability of bioactive substances and the speed of catalytic processes. Nanotechnology can be used in many areas of food production; for some, however, the methods are impractical and costly at an industrial scale. In addition, certain limitations (e.g., concentrations up to 100 μg mL^−1^) related to their use must be taken into account.

## Figures and Tables

**Figure 1 materials-16-00780-f001:**
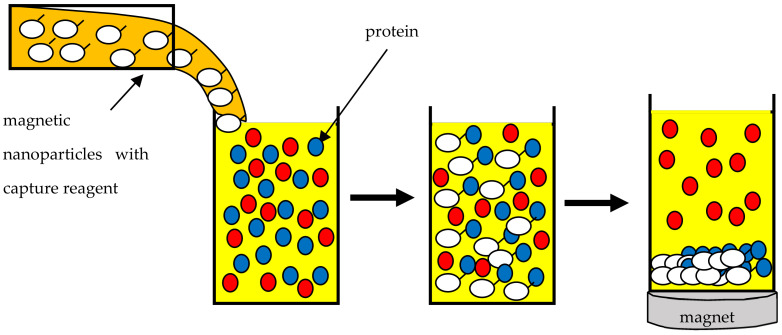
Schematic illustration of a protein purification system based on magnetic nanoparticles.

**Figure 2 materials-16-00780-f002:**
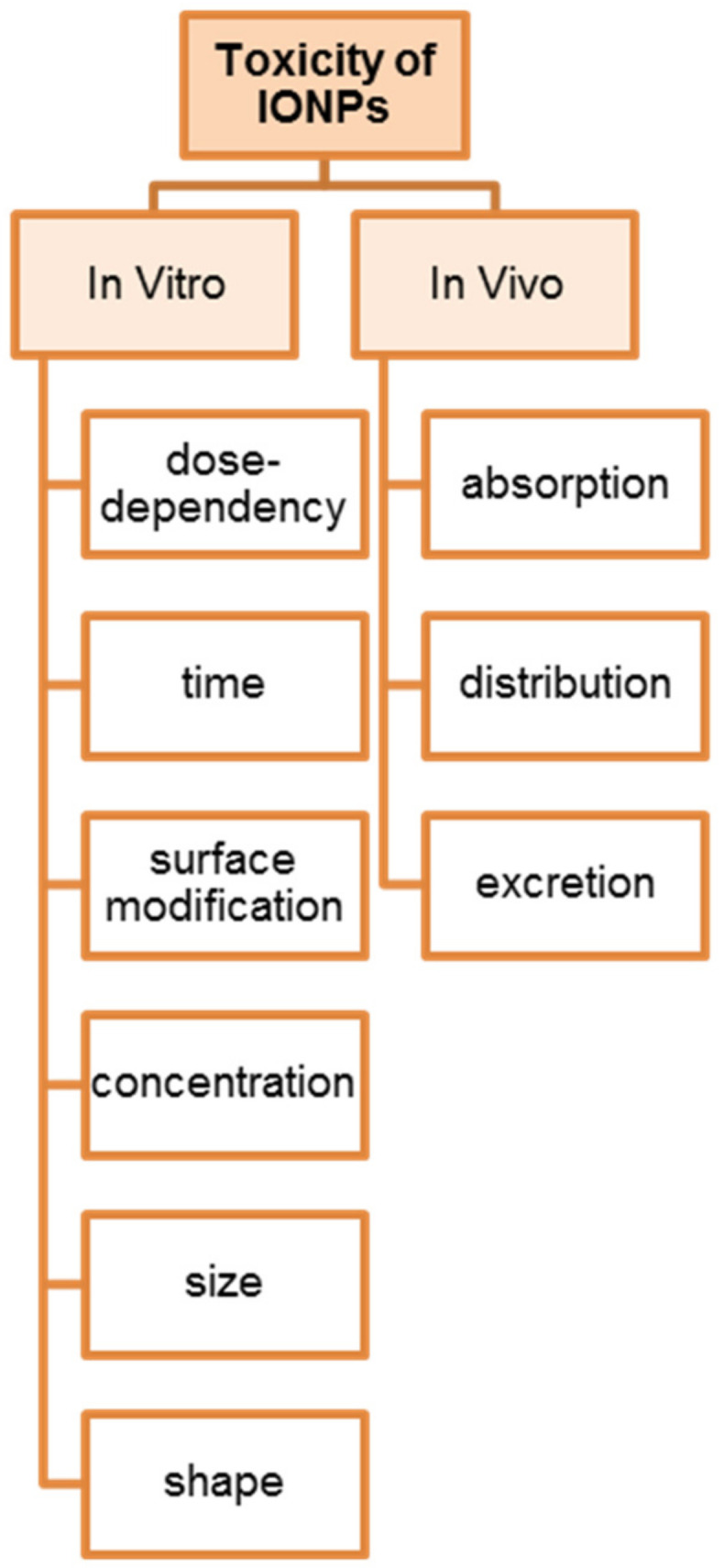
Criteria for evaluation of toxicity of IONPs.

**Table 1 materials-16-00780-t001:** Some typical synthesis methods for iron nanoparticles.

Method	Type	Shape	Size	References
Chemical reduction method	α-Fe	spherical	21 nm	[9]
	Fe^0^	spherical	10–80 nm	[10]
Co-precipitation	Fe_3_O_4_ and FeO	different	8.5 nm	[11]
	Fe_3_O_4_	-	20–22 nm	[12]
	Fe_3_O_4_	spherical	15 nm	[13]
	γ-Fe_2_O_3_	spherical	11.40 and 7.3657 nm	[14]
	γ-Fe_2_O_3_	spherical	from 5 to 8 nm	[15]
	Fe_3_O_4_	-	13.9 nm	[16]
	FeOH	polygonal disc	6.60 nm	[17]
Microemulsion synthesis	Fe_3_O_4_	spherical	~12 nm	[18]
	γ-Fe_2_O_3_	spherical	>10 nm	[19]
	Fe_3_O_4_	-	8–16 nm	[20]
Thermal decomposition	Fe and Fe_3_O_4_	spherical	5.41 and 7.38 nm	[21]
	Superparamagnetic iron oxide		2–30 nm	[22]
	Fe_3_O_4_		11 nm	[23]
	Fe_1−x_O@Fe_3−x_O_4_	cubes	17.5 nm length 6.3, nm thickness	[24]
Hydrothermal synthesis	α-Fe_2_O_3_ or Fe_3_O_4_	cubic and spherical	25 or 14 nm, respectively	[25]
	α-Fe_2_O_3_	spherical	15.6 nm	[26]
	FeO	rhombohedral	22, 14, 8 nm	[27]
Sonochemical process	α-Fe_2_O_3_	rhombohedral	From ~12 nm to ~19 nm	[28]
	Fe_3_O_4_	-	20 nm	[29]
Non-aqueous method	γ-Fe_2_O_3_	acicular shaped	major axis: 17 nm; minor axis: 1.7 nm	[30]
Sol–gel method	Fe_3_O_4_	spherical	79.04 nm	[31]
	α-Fe_2_O_3_	spherical and spheroid-shaped	12.7 nm	[32]

**Table 2 materials-16-00780-t002:** Examples of microorganisms used to synthesize iron nanoparticles.

Microorganism	Species	NPs	Precursor	References
Bacteria	*Actinobacter* sp.	γ-Fe_2_O_3_ maghemite Fe_3_O_4_ magnetite	aqueous potassium ferricyanide/ferrocyanide K_3_[Fe(CN)_6_]/ K_4_[Fe(CN)_6_]	[38]
	*Actinobacter* sp.	maghemite (γ-Fe_2_O_3_) and greigite (Fe_3_S_4_)	aqueous solution ferrous salts	[37]
	*Thermoanaerobacter* sp.	Fe_3_O_4_ magnetite	FeOOH	[39]
	*Bacillus subtilis*	Fe_3_O_4_	Fe_2_O_3_	[40]
	*Thiobacillus thioparus*	Fe_3_O_4_ magnetite	FeSO_4_	[41]
	*Alcaligenes faecalis*	Fe or their oxides	Fe_2_O_3_FeSO_4_	[42]
Fungi	*Pochonia chlamydosporium* *Aspergillus fumigatus* *Aspergillus wentii* *Curvularia lunata Chaetmium globosum*	Fe or their oxides	Fe_2_O_3_ FeSO_4_	[42]
	*Alternaria alternata*	γ-Fe_2_O_3_ and α-Fe_2_O_3_	FeCl_3_	[43]
	*Trichoderma asperellum* *Phialemoniopsis ocularis* *Fusarium incarnatum*	Fe oxides	FeCl_3_ FeCl_2_	[44]
Algae	*Sargassum muticum*	Fe_3_O_4_	ferric chloride solution	[45]
	*Chlorococcum* sp.	Fe	iron chloride	[46]

**Table 3 materials-16-00780-t003:** Examples of plants used to synthesize iron nanoparticles.

Part of Plant	Species	Precursor	References
Leaves	*Quercus* spp.	iron(III) chloride solution	[47]
	*Coriandrum sativum*	Ferric Chloride	[48]
	*Eucalyptus* spp.	FeSO_4_ and Ni(NO_3_)_2_	[49]
	*Moringa oleifera*	Iron nitrate(III) (Fe(NO_3_)_3_·9H_2_O)	[50]
	*Rosmarinus officinalis*	FeSO_4_	[51]
	*Daphne mezereum*	FeCl_3_·6H_2_O	[52]
	*Cassia fistula*	Fe_2_O_3_	[53]
Fruits	*Piper* spp.	FeCl_2_ and K_2_PdCl_4_	[54]
	*Terminalia bellirica* *Moringa oleifera*	ferric chloride heptahydrate (FeCl_2_·7H_2_O) and potassium ferricyanide (K_3_Fe(CN)_6_)	[55]
Peel	*Artocarpus heterophyllus*	FeCl_2_	[56]
	*Garcinia mangostana*	iron(II) chloride tetrahydrate (FeCl_2_·4H_2_O ≥ 99%) and iron(III) chloride hexahydrate (FeCl_3_·6H_2_O, 97%)	[57]
Flower	*Hibiscus sabdariffa*	FeCl_3_	[58]
Seeds	*Punica granatum*	iron chloride	[59]
	*Trigonella foenum-graecum*	FeCl_3_·6H_2_O	[60]
	*Syzygium cumini*	FeCl_3_·6H_2_O	[61]
Roots	*Chromolaena odorata*	solution of Fe(II) and Fe(III)	[62]
	*Zingiber officinale*	ferric chloride FeCl_3_	[63]
Brans	*Sorghum moench*	FeCl_3_	[64]
Buds	*Syzygium aromaticum*	iron chloride tetra-hydrate	[65]

**Table 4 materials-16-00780-t004:** Examples of synthesis of iron nanoparticles using green reagents.

Type of Reagent	Material	Precursor	Nanoparticles	References
Biopolymer	Starch	FeSO_4_·7H_2_O	Fe_3_O_4_	[68]
	Starch	FeCl_3_	Fe-Pd nanoparticles	[69]
	Sodium alginate	FeCl_3_	Fe_3_O_4_	[70]
	Agar	FeCl_2_·4H_2_O FeCl_3_·6H_2_O	Fe_3_O_4_	[71]
Polyphenol	Proanthocyanidin	ferric chloride solution	PACFeNPs	[70]
Acid	Ascorbic acid	FeCl_2_·4H_2_O and 2 mM of FeCl_3_·6H_2_O	SPION	[72]
	Aspartic acid	ferrous chloride tetra-hydrate (FeCl_2_·H_2_O), ferric chloride hexa-hydrate (FeCl_3_·6H_2_O)	A-IONPs	[73]
	Palmitic acid	rust	Fe_3_O_4_	[74]
	L-glutamic acid L-glutamine L-arginine L-cysteine	FeSO_4_·7H_2_O	nZVI	[75]
Oil	Sunflower oil	FeCl_3_·6H_2_O	nZVI	[76]

**Table 5 materials-16-00780-t005:** Examples of the use of IONPs in food analysis.

Food Type	Analyzed Substance/Organism	Material for Functionalized IONPs	References
Thai food	Cu(II)	chitosan–graphene quantum dots	[106]
milk and honey	sulfamethoxazole	molecularly imprinted polymers	[107]
fish, shrimp, canned tuna	Hg(II)	SiO_2_@polythiophene	[108]
cantaloupe, apple, nectarine	Cd(II), Cu(II), Ni(II)	2-aminobenzothiazole	[109]
milk	melamine	SiO2@MIPs	[110]
vegetable oil	pesticide	polystyrene	[111]
canned tuna fish, canned tomato paste, parsley, milk	Pb(II)	3-aminopropyl-trie- thoxysilane@ phthalic anhydride	[112]
candies and beverages	synthetic colorants	polyethyleneimine	[113]
sausage	nitrate	Au@l-cysteine	[114]
milk	*Salmonella typhimurium*	CdSe/ZnS QDs with NH 2−PEG−CM(MW 3400) as spacers to be conjugated withantibodies	[115]
milk	*Listeria monocytogenes*	carboxyl with rabbit anti-*Listeria* monocytogenes	[116]

**Table 6 materials-16-00780-t006:** The ways in which IONPs interact with microorganisms.

IONPs	Microorganism	Species	Effect	References
Fe_3_O_4_/SiO_2_ vancomycin	bacteria	*Bacillus cereus* *Staphylococcus aureus Shigella boydii* *Escherichia coli*	inhibition of bacterial growth	[18]
Fe^0^	bacteria	*Escherichia coli*	inactivation	[10]
		*Bacillus subtilis* *Escherichia coli* *Staphylococcus epidermidis*	3% provided full inhibition of microbial growth	[86]
		*Bacillus subtilis* *var. niger* *Pseudomonas fluorescens*	inactivation	[138]
	fungus	*Aspergillus versicolor*	inactivation	
		*Geotrichum candidum* *Rhodotorula rubra*	3% provided full inhibition of microbial growth	[86]
S-Fe^0^	bacteria	*Pseudomonas* spp. HLS-6 *Escherichia coli* DH5α	oxidative stress in the bacteria, destroys the cell structure and damages the intracellular DNA	[139]
IONzymy	viruses	Influenza A	induces peroxidation of membrane lipids in synthesized liposomes, inactivates viruses	[101]
Fe_3_O_4_	bacteria	*Escherichia coli* BW 25113 *Enterococcus hirae* ATCC 9790	change in membrane permeability, bactericidal effect on *E. coli*	[140]
	protozoan	*Plasmodium falciparum*	inhibited replication by inhibiting E protein expression	[141]
	virus	serotype DEN-2		
Superparamagnetic iron oxide	bacteria	*Pseudomonas aeruginosa* PA01	disintegration of the bacterial cell membrane by hyperthermia	[142]
		*Pseudomonas fluorescens*		[143]
γ-Fe_2_O_3_	fungus	*Phanerochaete chrysosporium*	concentration and exposure time affected malondialdehyde content, reactive oxygen species production and lactate dehydrogenase (LDH) activity	[144]
γ-Fe_2_O_3_ and α- Fe_2_O_3_	bacteria	*Escherichia coli* *Pseudomonas aeruginosa* *Bacillus subtilis* *Staphylococcus* *aureus*	inhibition of bacterial growth	[43]
IONPs	bacteria	*Staphylococcus aureus* *Staphylococcus epidermidis* *Escherichia coli* *Salmonella typhi* *Vibrio cholera*		[145]
	fungus	*Trichothecium roseum, Cladosporium herbarum, Penicillium chrysogenum* *Alternaria alternate* *Aspergillus niger*	inhibitory effect on growth	[146]
		*Aeromonas hydrophila* (ATCC 49140*)* *Aeromonas hydrophila* (MTCC 1739) *Aeromonas sobria* (MTCC 3613) *Aeromonas hydrophila*		[147]
FeOH	bacteria	*Staphylococcus aureus*	excellent antimicrobial activity	[17]
FeO-NPs/FeO-NRs	bacteria	*Staphylococcus aureus* *Escherichia coli* *Pseudomonas aeruginosa, Shigella* *Salmonella typhi* *Pasteurella*	at 10 μg mL^−1^, it showed a zone of growth inhibition	[148]

## Data Availability

Data sharing is not applicable to this article.

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
