# Peer review of "Application of Iron Nanoparticle-Based Materials in the Food Industry"

_materials, 2023, doi:10.3390/ma16020780_

Round 1

Reviewer 1 Report

This paper describes methods for obtaining and using 15 iron nanoparticles in food packaging, including edible coatings, artificial enzyme production, en-16 zyme immobilization, purification of proteins and foods from toxic substances, food analysis, pro-17 cess control, and food fortification. The second part of the article describes the biocompatibility of 18 iron nanoparticles, their effects on the human body and the safety of use.

Few suggestions are there to be addressed;

Please explain all the abbreviations at first instance of there use in the manuscript or it is better to provide list of abbreviations. e.g. TMAOH

Table 1, replace "Thermal decomposing" with "Thermal decomposition"

Page 3, Line 87, confirm the formula of hydrazine hydrate. Similarly other formulae are needed to be checked particularly where water of hydration is involved.

Table 2, Row 2nd, correct the spellings of "aqueous"

Table 4, Row 2nd, please check spelling of "Biopolymer"

Overall, the manuscript need thorough revision with respect to typo mistakes. Add some recent and relevant studies. Few are suggested;

Zeitschrift für Physikalische Chemie 235(8), 1055-1075, 2021

Zeitschrift für Physikalische Chemie, 233(9), 1325-1349, 2019

Author Response

The authors would like to thank the reviewer for his valuable comments.

Response to comments:

Please explain all the abbreviations at first instance of there use in the manuscript or it is better to provide list of abbreviations. e.g. TMAOH

The list of abbreviations and their explanations was introduced at the beginning of the manuscript

Table 1, replace "Thermal decomposing" with "Thermal decomposition"

It has been corrected.

Page 3, Line 87, confirm the formula of hydrazine hydrate. Similarly other formulae are needed to be checked particularly where water of hydration is involved.

It has been corrected.

Table 2, Row 2nd, correct the spellings of "aqueous"

It has been corrected.

Table 4, Row 2nd, please check spelling of "Biopolymer"

It has been corrected.

Overall, the manuscript need thorough revision with respect to typo mistakes. Add some recent and relevant studies. Few are suggested;

Zeitschrift für Physikalische Chemie 235(8), 1055-1075, 2021

Zeitschrift für Physikalische Chemie, 233(9), 1325-1349, 2019

The manuscript has been revised. The recommended works have been included in the manuscript.

Reviewer 2 Report

The manuscript of “Application of iron nanoparticle-based materials in the food industryis interesting to read and appreciated for good attempt. Author explained about several things like, iron nanoparticles in food production. I decided minor Revision and also correct the following suggestion before accept the manuscript.

1.      The abstract did not fulfill all the results and it should be written corrected with obtained results.

2.      Keywords should be in order and relate to the manuscript

3.      Hypothesis of introduction part is poor. Need proper explanation.

4.      In result part lot of general issues are discussed, images quality is low and the very normal.

5.      Som many references are old like ten years ago, try t to add new references.

Author Response

We would like to thank the reviewer for taking the time and pointing out legitimate comments on our work.

Response to comments:

The abstract did not fulfill all the results and it should be written corrected with obtained results.

The abstract has been rewritten.

Keywords should be in order and relate to the manuscript

Keywords have been corrected.

Hypothesis of introduction part is poor. Need proper explanation.

It was revised and completed.

In result part lot of general issues are discussed, images quality is low and the very normal.

The result section discusses many general issues, the quality of the images is low and very normal. The specificity of the article touching on many aspects of the use of materials based on iron nanoparticles sometimes required citing the issue in a general way. Deeper analysis would have led to overextension of the manuscript. The literature cited makes it possible to reach detailed information.

The quality of the drawings has been improved.

Some many references are old like ten years ago, try t to add new references.

The comment has been partially accepted. Wherever possible, citations have been replaced with newer ones. Some works describe methods that are not used in newer works or the appropriate analysis has not been done. Hence, the authors believe that it is better to cite a source paper than a more recent paper in which this source paper was described. Such works in the entire manuscript are approx. 10%. The remaining bibliography is more recent than 10 years.

Reviewer 3 Report

The manuscript is a review article on the application of iron nanoparticle-based materials in the food industry. The article is comprehensive, and the topic is interesting. The authors studied many literature sources, and the researchers' results are presented in the manuscript.

The manuscript has some weaknesses related to the organization of the article. The authors should create a new chapter for the application of iron nanoparticles in the food industry, current chapters 3, 5-15 should be subchapter in new chapter. Chapter 4 may be incorporated into the second part of the article which describe the biocompatibility of iron nanoparticles, their effects on the human body and the safety of use.

Line 63; the abbreviation TMAOH should be explain.

Line 67; it is not clear what authors mean with CO (carbonmonoxy?)

Write 0 in superscript when writing zero valent Fe, Table 1, line 157, line 159, line 200 write 0 superscript when writing zero valent Fe

Write subscript NaBH4 line 88, table 3, H2O

Line 87, formula of hydrazone hydrate is not correct.

Line 91, correct FeCl3-6H2O and FeSO4-7H2O, delete – and write ×.

Line 129-131 sentence: In addition, the synthesis of nano-particles in an aqueous environment and under standard conditions (temperature and pressure) excludes the formation of toxic by-products and eliminates organic waste disposal problems. Which toxic by product and eliminates organic waste disposal problems? The authors should clarify

It is not clear what Figure 1 is supposed to show

Write space between number and the unit line 216

Line 284; explain what IONzyme is.

Line 299; explain what the protein A is.

Line 300; explain what the Ara h3/4 is.

Delete space between chemical symbols and valence number example Pb(II) and not Pb (II) corrected in the whole document and also deleted space between iron (II), it need to be iron(II) corrected in the whole document.

In Table 4, error writing Fe3O3

The last sentence in the summary needs further explanation, the use of nanoparticles has not only advantages but also disadvantages. We need to be aware of the disadvantages, especially those that affect people's health.

Author Response

I am sending the revised version, which has been greatly improved thanks to your valuable comments. Thank you very much for taking the time to review so that I can provide a better version of the submitted manuscript, and please consider responding to the comments in the review.

The manuscript has some weaknesses related to the organization of the article. The authors should create a new chapter for the application of iron nanoparticles in the food industry, current chapters 3, 5-15 should be subchapter in new chapter. Chapter 4 may be incorporated into the second part of the article which describe the biocompatibility of iron nanoparticles, their effects on the human body and the safety of use.

Suggested changes were made to the manuscript.

Line 63; the abbreviation TMAOH should be explain.

The abbreviation TMAOH is explained in the chapter “Abbreviations” on the first page of the article.

Line 67; it is not clear what authors mean with CO (carbonmonoxy?)

As reported in the literature, magnetite/hematite nanocomposites and a pure phase of magnetite nanoparticles were obtained by thermal decomposition of hematite powder (α-Fe2O3) in the presence of a high boiling point solvent in the presence of CO or H2 to inhibit the effect of oxygen. Information on the use of CO for reduction has been removed.

Write 0 in superscript when writing zero valent Fe, Table 1, line 157, line 159, line 200 write 0 superscript when writing zero valent Fe

It has been corrected

Write subscript NaBH4 line 88, table 3, H2O

It has been corrected. In addition, other incorrectly written values have been corrected.

Line 87, formula of hydrazone hydrate is not correct.

It has been corrected.

Line 91, correct FeCl3-6H2O and FeSO4-7H2O, delete – and write ×.

It has been corrected.

Line 129-131 sentence: In addition, the synthesis of nano-particles in an aqueous environment and under standard conditions (temperature and pressure) excludes the formation of toxic by-products and eliminates organic waste disposal problems. Which toxic by product and eliminates organic waste disposal problems? The authors should clarify

The statement on the generation of toxic waste and the disposal of organic waste has been clarified.

It is not clear what Figure 1 is supposed to show

Figure 1 has been removed.

Write space between number and the unit line 216

It has been corrected.

Line 284; explain what IONzyme is.

IONzyme is explained in the chapter “Abbreviations” on the first page of the article.

Line 299; explain what the protein A is.

It has been explained.

Line 300; explain what the Ara h3/4 is.

Ara h3/4 has been changed to Ara h3 and explained.

Delete space between chemical symbols and valence number example Pb(II) and not Pb (II) corrected in the whole document and also deleted space between iron (II), it need to be iron(II) corrected in the whole document.

Space between chemical symbols and valence number have been removed throughout the document.

In Table 4, error writing Fe3O3

Fe3O3 and Fe3O4 in table 4 have been replaced by nZVI as referenced in the relevant publication.

The last sentence in the summary needs further explanation, the use of nanoparticles has not only advantages but also disadvantages. We need to be aware of the disadvantages, especially those that affect people's health.

The last sentence in the summary was extended by advantages and disadvantages.

Reviewer 4 Report

 Manuscript is devoted to summarizing the applications of iron nanoparticles in food production and analyzes  the safety of their consumption by humans. Methods for obtaining iron nanoparticles such as chemical synthesis, green synthesis (synthesis by microorganisms), synthesis using green reagents are considered.   Application in the production of food packaging is discussed. Cellular mechanisms of toxicity of iron nanoparticles and antimicrobial effects of Iron oxide nanoparticles in relation to bacteria, viruses, fungus, protozoan, etc. are considered. Immobilization of enzymes and artificial enzymes are considered. The possibility of protein purification and Iron oxides as ingredients in foods and dietary supplements and control of the process flow and anti-allergic effect are analyzed. Particular attention is paid to the criteria for evaluation of toxicity of Iron oxide nanoparticles and their impact on the human body. The authors concluded that the application of iron nanoparticles in the food sector undoubtedly contributes to the development of food technology. The manuscript may be interesting and useful to a wide range of readers.

Author Response

Thank you for your input and taking the time to review our article. The attached comments do not require comment.

Round 2

Reviewer 3 Report

The authors have taken into account all the comments and improved the article. In my opinion, the article is suitable for publication.